# Learning Search Space Partition for Black-box Optimization using Monte Carlo Tree Search

**Linnan Wang**
Brown University
linnan_wang@brown.edu

**Rodrigo Fonseca**
Brown University
rfonseca@cs.brown.edu

**Yuandong Tian**
Facebook AI Research
yuandong@fb.com

## Abstract

High dimensional black-box optimization has broad applications but remains a challenging problem to solve. Given a set of samples $\{\mathbf{x}_i, y_i\}$, building a global model (like Bayesian Optimization (BO)) suffers from the curse of dimensionality in the high-dimensional search space, while a greedy search may lead to sub-optimality. By recursively splitting the search space into regions with high/low function values, recently LaNAS [1] shows good performance in Neural Architecture Search (NAS), reducing the sample complexity empirically. In this paper, we coin *LA-MCTS* that extends LaNAS to other domains. Unlike previous approaches, LA-MCTS *learns* the partition of the search space using a few samples and their function values in an online fashion. While LaNAS uses linear partition and performs uniform sampling in each region, our LA-MCTS adopts a nonlinear decision boundary and learns a local model to pick good candidates. If the nonlinear partition function and the local model fit well with ground-truth black-box function, then good partitions and candidates can be reached with much fewer samples. LA-MCTS serves as a *meta-algorithm* by using existing black-box optimizers (e.g., BO, TuRBO [2]) as its local models, achieving strong performance in general black-box optimization and reinforcement learning benchmarks, in particular for high-dimensional problems.

## 1   Introduction

Black-box optimization has been extensively used in many scenarios, including Neural Architecture Search (NAS) [3, 1, 4], planning in robotics [5, 6], hyper-parameter tuning in large scale databases [7] and distributed systems [8], integrated circuit design [9], etc.. In black-box optimization, we have a function $f$ without explicit formulation and the goal is to find $\mathbf{x}^*$ such that

$$\mathbf{x}^* = \arg \max_{\mathbf{x} \in X} f(\mathbf{x}) \tag{1}$$

with the fewest samples ($\mathbf{x}$). In this paper, we consider the case that $f$ is deterministic.

Without knowing any structure of $f$ (except for the local smoothness such as Lipschitz-continuity [10]), in the worst-case, solving Eqn. 1 takes exponential time, i.e. the optimizer needs to search every $\mathbf{x}$ to find the optimum. One way to address this problem is through *learning*: from a few samples we *learn* a surrogate regressor $\hat{f} \in \mathcal{H}$ and optimize $\hat{f}$ instead. If the model class $\mathcal{H}$ is small and $f$ can be well approximated within $\mathcal{H}$, then $\hat{f}$ is a good approximator of $f$ with much fewer samples.

Many previous works go that route, such as Bayesian Optimization (BO) and its variants [11, 12, 13, 14]. However, in the case that $f$ is highly nonlinear and high-dimensional, we need to use a very large model class $\mathcal{H}$, e.g. Gaussian Processes (GP) or Deep Neural Networks (DNN), that requires many samples to fit before generalizing well. For example, Oh et al [15] observed that the myopic acquisition in BO over-explores the boundary of a search space, especially in high dimensional problems. To address this issue, recent works start to explore space partitioning [5, 16, 17] and local

modeling [2, 18] that fits local models in promising regions, and achieve strong empirical results in high dimensional problems. However, their space partitions follow a fixed criterion (e.g., $K$-ary uniform partition) that is independent of the objective to be optimized.

Following the path of learning, one under-explored direction is to *learn the space partition*. Compared to learning a regressor $\hat{f}$ that is expected to be accurate in the region of interest, it suffices to learn a classifier that puts the sample to the right subregion with high probability. Moreover, its quality requirement can be further reduced if done recursively.

In this paper, we propose LA-MCTS, a meta-level algorithm that recursively learns space partition in a hierarchical manner. Given a few samples within a region, it first performs unsupervised $K$-mean algorithm based on their function values, learns a classifier using $K$-mean labels, and partition the region into good and bad sub-regions (with high/low function value). To address the problem of mis-partitioning good data points into bad regions, LA-MCTS uses UCB to balance exploration and exploitation: it assigns more samples to good regions, where it is more likely to find an optimal solution, and exploring other regions in case there are good candidates. Compared to previous space partition method, e.g. using Voronoi graph [5], we learn the partition that is adaptive to the objective function $f(\mathbf{x})$. Compared to the local modeling method, e.g. TuRBO [2], our method dynamically exploits and explores the promising region w.r.t samples using Monte Carlos Tree Search (MCTS), and constantly refine the learned boundaries with new samples.

LA-MCTS extends LaNAS [1] in three aspects. First, while LaNAS learns a hyper-plane, our approach learns a non-linear decision boundary that is more flexible. Second, while LaNAS simply performs uniform sampling in each region as the next sample to evaluate, we make the key observation that local model works well and use existing solvers such as BO to find a promising data point. This makes LA-MCTS a *meta-algorithm* usable to boost existing algorithms that optimize via building local models. Third, while LaNAS mainly focus on Neural Architecture Search (< 20 discrete parameters), our approach shows strong performance on generic black-box optimization.

We show that LA-MCTS, when paired with TurBO, outperforms various SoTA black-box solvers from Bayesian Optimizations, Evolutionary Algorithm, and Monte Carlo Tree Search, in several challenging benchmarks, including *MuJoCo locomotion tasks*, trajectory optimization, reinforcement learning, and high-dimensional synthetic functions. We also perform extensive ablation studies, showing LA-MCTS is relatively insensitive to hyper-parameter tuning. As a meta-algorithm, it also substantially improves the baselines.

The implementation of LA-MCTS can be found at https://github.com/facebookresearch/LaMCTS.

## 2  Related works

Bayesian Optimization (BO) has become a promising approach in optimizing the black-box functions [11, 12, 13], despite much of its success is typically limited to less than 15 parameters [19] and a few thousand evaluations [18]. While most real-world problems are high dimensional, and reliably optimizing a complex function requires many evaluations. This has motivated many works to scale up BO, by approximating the expensive Gaussian Process (GP), such as using Random Forest in SMAC [20], Bayesian Neural Network in BOHAMIANN [21], and the tree-structured Parzen estimator in TPE [22]. BOHB [23] further combines TPE with Hyperband [24] to achieve strong any time performance. Therefore, we choose BOHB in comparison. Using a sparse GP is another way to scale up BO [25, 26, 27]. However, sparse GP only works well if there exists sample redundancy, which is barely the case in high dimensional problems. Therefore, scaling up evaluations is not sufficient for solving high-dimensional problems.

There are lots of work to specifically study high-dimensional BO [28, 29, 30, 31, 32, 33, 34, 35, 36, 37]. One category of methods decomposes the target function into several additive structures [32, 35], which limits its scalability by the number of decomposed structures for training multiple GP. Besides, learning a good decomposition remains challenging. Another category of methods is to transform a high-dimensional problem in low-dimensional subspaces. REMBO [34] fits a GP in low-dimensional spaces and projects points back to a high-dimensional space that contains the global optimum with a reasonable probability. Binois et al [38] further improves the distortion from Gaussian projections in REMBO. While REMBO works empirically, HesBO [19] is a theoretical sound framework for BO that optimizes high-dimensional problems on low dimensional sub-spaces embeddings; In BOCK [15],

Table 1: Definition of notations used through this paper.

| $\mathbf{x}_i$ | the ith sample | $f(\mathbf{x}_i)$ | the evaluation of $\mathbf{x}_i$ | $D_t$ | collected $\{\mathbf{x}_i, \mathrm{f}(\mathbf{x}_i)\}$ from iter $1 \rightarrow t$ |
|---|---|---|---|---|---|
| $\Omega$ | the entire search space | $\Omega_j$ | the partition represented by node $j$ | $D_t \cap \Omega_j$ | samples classified in $\Omega_j$ |
| $n_j$ | #visits at node j | $v_j$ | the value of node j | $ucb_j$ | the ucb score of node j |

Oh et al observed existing BO spends most evaluations near the boundary of a search space due to the Euclidean geometry, and it proposed transforming the problem into a cylindrical space to avoid over-exploring the boundary. EBO [18] uses an ensemble of local GP on the partitioned problem space. Based on the same principle of local modeling as EBO, recent trust-region BO (TuRBO) [2] has outperformed other high-dimensional BO on a variety of tasks. In comparing to high dimensional BO, we picked SoTA local modeling method TuRBO and dimension reduction method HesBO.

Evolutionary Algorithm (EA) is another popular algorithm for high dimensional black-box optimizations. A comprehensive review of EA can be found in [39]. CMA-ES is a successful EA method that uses co-variance matrix adaption to propose new samples. Differential Evolution (DE) [40] is another popular EA approach that uses vector differences for perturbing the vector population. Recently, Liu et al proposes a metamethod (Shiwa) [41] to automatically selects EA methods based on hyper-parameters such as problem dimensions, budget, and noise level, etc., and Shiwa delivers better empirical results than any single EA method. We choose Shiwa, CMA-ES, and differential evolution in comparisons.

Besides the recent success in games [42, 43, 44, 45], Monte Carlo Tree Search (MCTS) is also widely used in the robotics planning and optimization [6, 46, 47, 48]. Several space partitioning algorithms have been proposed in this line of research. In [16], Munos proposed DOO and SOO. DOO uses a tree structure to partition the search space by recursively bifurcating the region with the highest upper bound, i.e. optimistic exploration, while SOO relaxes the Lipschitz condition of DOO on the objective function. HOO [14] is a stochastic version of DOO. While prior works use K-ary partitions, Kim et al show Voronoi [5] partition can be more efficient than previous linear partitions in high-dimensional problems. In this paper, based on the idea of space partitioning, we extend current works by learning the space partition so that the partition can adapt to the distribution of $f(\mathbf{x})$. Besides, we improve the sampling inside a selected region with BO. This also helps BO from over-exploring by bounding within a small region.

## 3 Methodology

### 3.1 Latent Action Monte Carlo Tree Search (LA-MCTS)

This section describes LA-MCTS that progressively partitions the problem space. Please refer to Table 1 for definitions of notations in this paper.

**The model of MCTS search tree**: At any iteration t, we have a dataset $D_t$ collected from previous evaluations. Each entry in $D_t$ contains a pair of $(\mathbf{x}_i, f(\mathbf{x}_i))$. A tree node (e.g. node A in Fig. 1) represents a region $\Omega_A$ in the entire problem space ($\Omega$), then $D_t \cap \Omega_A$ represents the samples falling within node A. Each node also tracks two important statistics to calculate UCB1 [49] for guiding the selection: $n_A$ represents the number of visits at node A, which is the #sample in $D_t \cap \Omega_A$; and $v_i$ represents the node value that equals to $\frac{1}{n_i} \sum f(\mathbf{x}_i), \forall \mathbf{x}_i \in D_t \cap \Omega_i$.

LA-MCTS finds the promising regions by recursively partitioning. Starting from the root, every internal node, e.g. node A in Fig. 1, use *latent actions* to bifurcate the region represented by itself into a high performing and a low performing disjoint region ($\Omega_B$ and $\Omega_C$) for its left and right child, respectively (by default we use left child to represent a good region), and $\Omega_A = \Omega_B \cup \Omega_C$. Then a tree enforces the behavior of recursively partitioning from root

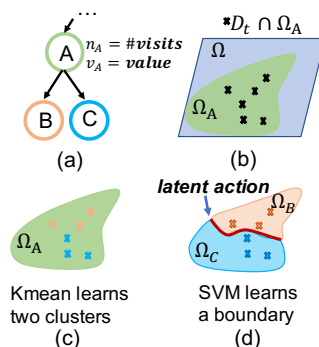

Figure 1: **The model of latent actions**: each tree nodes represents a region in the search space, and *latent action* splits the region into a high-performing and a low-performing region using $\mathbf{x}$ and $f(\mathbf{x})$.

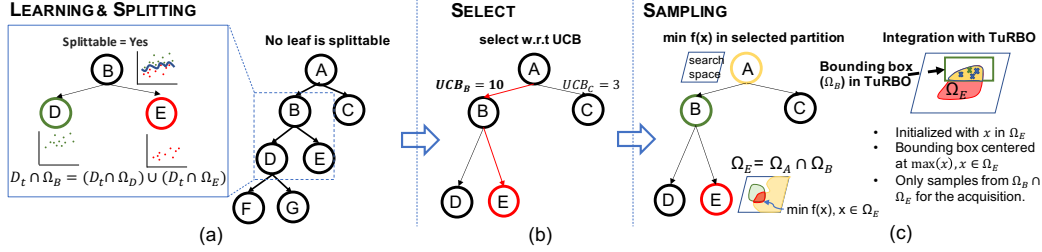

Figure 2: **The workflow of LA-MCTS**: In an iteration, LA-MCTS starts with building the tree via splitting, then it selects a region based on UCB. Finally, on the selected region, it samples by BO.

to leaves so that regions represented by tree leaves ($\Omega_{leaves}$) can be easily ranked from the best (the leftmost leaf), the second-best (the sibling of the leftmost leaf) to the worst (the rightmost leaf) due to the partitioning rule. The tree grows as the optimization progress, $\Omega_{leaves}$ becomes smaller, better focusing on a promising region (Fig. 7(b)). Please see sec 3.1.1 for the tree construction. By directly optimizing on $\Omega_{leaves}$, it helps BO from over-exploring, hence improving the BO performance especially in high dimensional problems.

**Latent actions**: Our model defines *latent action* as a boundary that splits the region represented by a node into a high-performing and a low performing region. Fig. 1 illustrates the concept and the procedures of creating latent actions on a node. Our goal is to learn a boundary from samples in $D_t \cap \Omega_A$ to maximize the performance difference of two regions split by the boundary. We apply Kmeans on the feature vector of $[\mathbf{x}, f(\mathbf{x})]$ to find a good and a bad performance clusters in $D_t \cap \Omega_A$, then use SVM to learn a decision boundary. Learning a nonlinear decision boundary is a traditional Machine Learning (ML) task, Neural Networks (NN) and Support Vector Machines (SVM) are two typical solutions. We choose SVM for the ease of training, and requiring fewer samples to generalize well in practices. Please note a simple node model is critical for having a tree of them. For the same reason, we choose Kmeans to find two clusters with good and bad performance. The detailed procedures are as follows:

1. At any node A, we prepare $\forall [\mathbf{x}_i, f(\mathbf{x}_i)], i \in D_t \cap \Omega_j$ as the training data for Kmeans to learn two clusters of different performance (Fig. 1 (b, c)), and get the cluster label $l_i$ for every $\mathbf{x}_i$ using the learned Kmeans, i.e. $[l_i, \mathbf{x}_i]$. So, the cluster with higher average f($\mathbf{x}_i$) represents a good performing region, and lower average f($\mathbf{x}_i$) represents a bad region.

2. Given $[l_i, \mathbf{x}_i]$ from the previous step, we learn a boundary with SVM to generalize two regions to unseen $\mathbf{x}_i$, and *the boundary learnt by SVM forms the latent action* (Fig. 1(d)). for example, $\forall \mathbf{x}_i \in \Omega$ with predicted label equals the high-performing region goes to the left child, and right otherwise.

### 3.1.1 The search procedures

Fig. 2 summarizes a search iteration of LA-MCTS that has 3 major steps. 1) *Learning and splitting* dynamically deepens a search tree using new $\mathbf{x}_i$ collected from the previous iteration; 2) *select* explores partitioned search space for sampling; and 3) *sampling* solves $minimize f(\mathbf{x}_i), \mathbf{x}_i \in \Omega_{selected}$ using BO, and SVMs on the selected path form constraints to bound $\Omega_{selected}$. We omit the back-propagation as it is implicitly done in splitting. Please see [4, 45] for a review of regular MCTS.

**Dynamic tree construction via splitting**: we estimate the performance of a $\Omega_i$, i.e. $v_i^*$, by $\hat{v}_i^* = \frac{1}{n_i} \sum f(\mathbf{x}_i), \forall \mathbf{x}_i \in D_t \cap \Omega_i$. At each iterations, new $\mathbf{x}_i$ are collected and the regret of $|\hat{v}_i^* - v_i^*|$ quickly decreases. Once the regret reaches the plateau, new samples are not necessary; then LA-MCTS splits the region using *latent actions* (Fig. 1) to continue refining the value estimation of two child regions. With more and more samples from promising regions, the tree becomes deeper into good regions, better guiding the search toward the optimum. In practice, we use a threshold $\theta$ as a tunable parameter for splitting. If the size of $D_t \cap \Omega_i$ exceeds the threshold $\theta$ at any leaves, we split the leaf with *latent actions*. We presents the ablation study on $\theta$ in Fig. 8.

The structure of our search tree dynamically changes across iterations, which is different from the pre-defined fixed-height tree used in LaNAS [1]. At the beginning of an iteration, starting from the

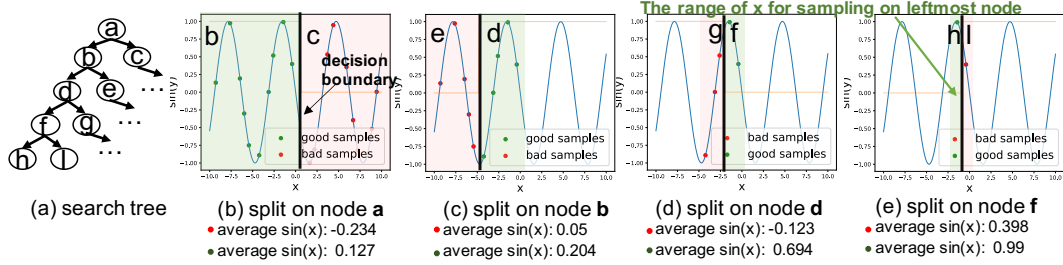

Figure 3: The visualization of partitioning 1d $\sin(x)$ using LA-MCTS.

root that contains all the samples, we recursively split leaves using *latent actions* if the sample size of any leaves exceeds the splitting threshold $\theta$, e.g. creating node D and node E for node B in Fig.2(a). We stop the tree splitting until no more leaves satisfy the splitting criterion. Then, the tree is ready to use in this iteration.

**Select via UCB**: According to the partition rule, a simple greedy based *go-left* strategy can be used to exclusively exploit the current most promising leaf. This makes the algorithm over-exploiting a region based on existing samples, while the region can be sub-optimal with the global optimum located in a different place. To build an accurate global view of $\Omega$, LA-MCTS selects a partition following Upper Confidence Bound (UCB) for the adaptive exploration; and the definition of UCB for a node is $ucb_j = \frac{v_j}{n_j} + 2C_p * \sqrt{2log(n_p)/n_j}$, where $C_p$ is a tunable hyper-parameter to control the extent of exploration, and $n_p$ represents #visits of the parent of node j. At a parent node, it chooses the node with the largest $ucb$ score. By following UCB from the root to a leaf, we select a path for sampling (Fig. 2(b)). When $C_p = 0$, UCB degenerates to a pure greedy based policy, e.g. *regression tree*. An ablation study on $C_p$ in Fig. 8(a) highlights that the exploration is critical to the performance.

**Sampling via Bayesian Optimizations**: *select* finds a path from the root to leaf, and SVMs on the path collectively intersects a region for sampling (e.g. $\Omega_E$ in Fig. 2(c)). In *sampling*, LA-MCTS solves $minf(\mathbf{x})$ on a constrained search space $\Omega_{selected}$, e.g. $\Omega_E$ in Fig. 2(c).

*Sampling with TuRBO*: here we illustrate the integration of SoTA BO method TuRBO [2] with LA-MCTS. We use TuRBO-1 (no bandit) for solving $minf(\mathbf{x})$ within the selected region, and make the following changes inside TuRBO, which is summarized in Fig. 2(c). a) At every re-starts, we initialize TuRBO with random samples only in $\Omega_{selected}$. The shape of $\Omega_{selected}$ can be arbitrary, so we use the rejected sampling (uniformly samples and reject outliers with SVM) to get a few points inside $\Omega_{selected}$. Since we only need a few samples for the initialization, the reject sampling is sufficient. b) TuRBO centers a bounding box at the best solution so far, while we restrict the center to be the best solution in $\Omega_{selected}$. c) TuRBO uniformly samples from the bounding box to feed the acquisition to select the best as the next sample, and we restrict the TuRBO to uniformly sample from the intersection of the bounding box and $\Omega_{selected}$. The intersection is guaranteed to exist because the center is within $\Omega_{selected}$. At each iteration, we keep TuRBO running until the size of trust-region goes 0, and all the evaluations, i.e. $\mathbf{x}_i$ and $f(\mathbf{x}_i)$, are returned to LA-MCTS to refine learned boundaries in the next iteration. Noted our method is also extensible to other solvers by following similar procedures.

*Sampling with regular BO*: following the steps described in Sec. 3.1.1, we select a leaf for sampling by traversing down from the root. The formulation of sampling with BO is same as using other solvers that $minf(\mathbf{x}), \mathbf{x} \in \Omega_{selected}$. $\Omega_{selected}$ is constrained by SVMs on the selected path. We optimize the acquisition function of BO by sampling, while sampling in a bounded arbitrary $\Omega_{selected}$ is non-trivial especially in high-dimensional space. For example, *rejected sampling* can fail to work as the search space is too large to get sufficient random samples in $\Omega_{selected}$; *hit-and-run* [50] or *Gibbs sampling* [51] can be good alternatives. In Fig. 4, we propose a new heuristic for sampling. At every existing samples $\mathbf{x}$ inside $\Omega_{selected}$, we draw a

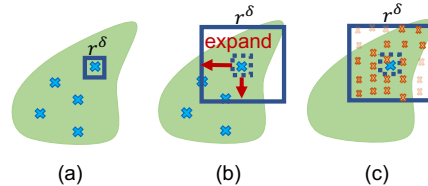

(a)     (b)     (c)

Figure 4: Illustration of sampling steps in optimizing the acquisition for Bayesian Optimization. We uniformly draw samples within a hyper-cube, then expand the cube and reject outliers.

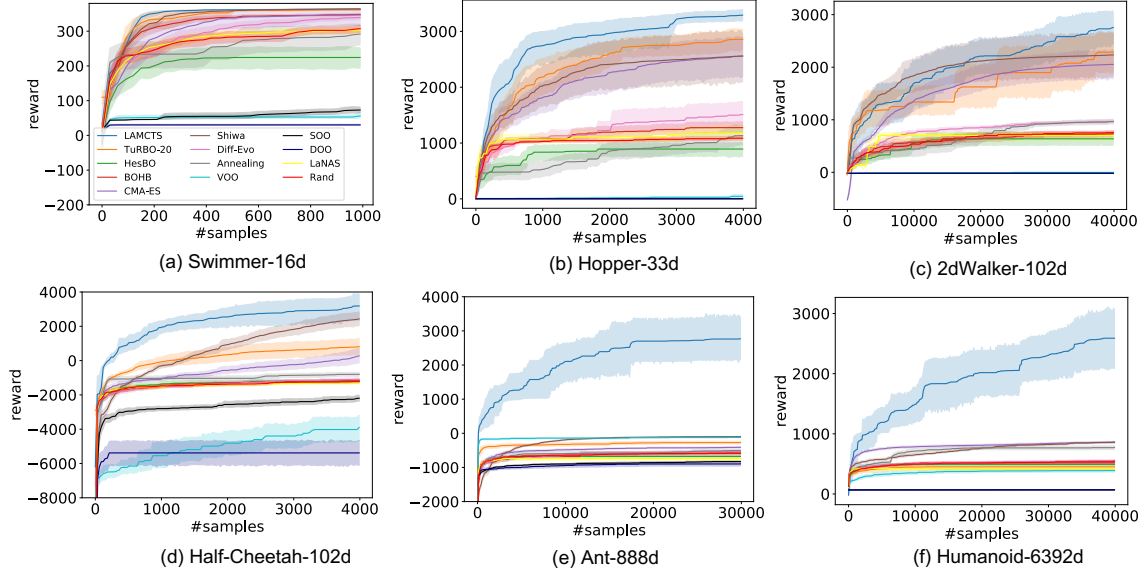

Figure 5: **Benchmark on MuJoCo locomotion tasks**: LA-MCTS consistently outperforms baselines on 6 tasks. With more dimensions, LA-MCTS shows stronger benefits (e.g. Ant and Humanoid). This is also observed in Fig. 6. Due to exploration, LA-MCTS experiences relatively high variance but achieves better solution after 30k samples, while other methods quickly move into local optima due to insufficient exploration.

rectangle $r^\delta$ of length equals to $\delta$ centered at $\mathbf{x}_i$ (Fig. 4(a)),
and $\mathbf{x}_i \in \Omega \cap D_t$, where $\delta$ is a small constant (e.g. $10^{-4}$). Next, we uniformly draw random samples using sobol sequence [52] inside $r^\delta$. Since $\delta$ is a small constant, we assume all the random samples located inside $\Omega_{selected}$. Then we linearly scale both the rectangle $r^\delta$ and samples within $r^\delta$ until certain percentages (e.g. 10) of samples located outside of $\Omega_{selected}$ (Fig. 4(b)). We keep those samples that located inside $\Omega_{selected}$ (Fig. 4(c)) for optimizing the acquisition, and repeat the procedures for every existing samples in $\Omega_{selected} \cap D_t$. Finally, we propose the sample with the largest value calculated from the acquisition function.

Fig. 3 (in Appendix) provides an example of partitioning 1d sin(x) using LA-MCTS.

## 4 Experiments

We evaluate LA-MCTS against the SoTA baselines from different algorithm categories ranging from Bayesian Optimization (TuRBO [2], HesBO [19], BOHB [23]), Evolutionary Algorithm (Shiwa [41], CMA-ES [53], Differential Evolution (DE) [40]), MCTS (VOO [5], SOO [16], and DOO [16]), Dual Annealing [54] and Random Search. In experiments, LA-MCTS is defaulted to use TuRBO for sampling unless state otherwise. For baselines, we used the authors' reference implementations (see the bibliography for the source of implementations). The hyper-parameters of baselines are optimized toward tasks and the setup of each algorithm can be found in Appendix A.1.

### 4.1 MuJoCo locomotion tasks

MuJoCo [55] locomotion tasks (*swimmer*, *hopper*, *walker-2d*, *half-cheetah*, *ant* and *humanoid*) are among the most popular Reinforcement Learning (RL) benchmarks, and learning a *humanoid* model is considered one of the most difficult control problems solvable by SoTA RL methods [56]. While the push and trajectory optimization problems used in [2, 18] only have up to 60 parameters, MuJoCo tasks are more difficult: e.g., the most difficult task *humanoid* in MuJoCo has 6392 parameters.

Here we chose the linear policy $\mathbf{a} = \mathbf{W}\mathbf{s}$ [57], where $\mathbf{s}$ is the state vector, $\mathbf{a}$ is the action vector, and $\mathbf{W}$ is the linear policy. To evaluate a policy, we average rewards from 10 episodes. We want to find $\mathbf{W}$ to maximize the reward. Each component of $\mathbf{W}$ is continuous and in the range of $[-1, 1]$.

Table 2: Compare with gradient-based approaches on MuJoCo v1; and the performance on MuJoCo v2 is similar. Despite being a black-box optimizer, LA-MCTS still achieves good sample efficiency in low-dimensional tasks (*Swimmer*, *Hopper* and *HalfCheetah*), but lag behind in high-dimensional tasks due to excessive burden in exploration, which gradient approaches lack.

| Task | Reward Threshold | The average episodes (#samples) to reach the threshold | | | | |
|---|---|---|---|---|---|---|
| | | LA-MCTS | ARS V2-t [57] | NG-lin [58] | NG-rbf [58] | TRPO-nn [57] |
| Swimmer-v2 | 325 | **126** | 427 | 1450 | 1550 | N/A |
| Hopper-v2 | 3120 | 2913 | 1973 | 13920 | 8640 | 10000 |
| HalfCheetah-v2 | 3430 | 3967 | 1707 | 11250 | 6000 | 4250 |
| Walker2d-v2 | 4390 | N/A($r_{best} = 3523$) | 24000 | 36840 | 25680 | 14250 |
| Ant-v2 | 3580 | N/A($r_{best} = 2871$) | 20800 | 39240 | 30000 | 73500 |
| Humanoid-v2 | 6000 | N/A($r_{best} = 3202$) | 142600 | 130000 | 130000 | unknown |

N/A stands for not reaching reward threshold.
$r_{best}$ stands for the best reward achieved by LA-MCTS under the budget in Fig. 5.

Fig. 5 suggests LA-MCTS consistently out-performs various SoTA baselines on all tasks. In particular, on high-dimensional hard problems such as *ant* and *humanoid*, the advantage of LA-MCTS over baselines is the most obvious. Here we use TuRBO-1 to sample $\Omega_{selected}$ (see sec. 3.1.1). **(a) vs TuRBO**. LA-MCTS substantially outperforms TuRBO: with learned partitions, LA-MCTS reduces the region size so that TuRBO can fit a better model in small regions. Moreover, LA-MCTS helps TuRBO initialize from a promising region at every restart, while TuRBO restarts from scratch. **(b) vs BO**. While BO variants (e.g., BOHB) perform very well in low-dimensional problem (Fig. 5), their performance quickly deteriorates with increased problem dimensions (Fig. 5(b)→(f)) due to over-exploration [15]. LA-MCTS prevents BO from over-exploring by quickly getting rid of unpromising regions. By traversing the partition tree, LA-MCTS also completely removes the step of optimizing the acquisition function, which becomes harder in high dimensions. **(c) vs objective-independent space partition**. Methods like VOO, SOO, and DOO use hand-designed space partition criterion (e.g., $k$-ary partition) which does not adapt to the objective. As a result, they perform poorly in high-dimensional problems. On the other hand, LA-MCTS learns the space partition that depends on the objective $f(\mathbf{x})$. The learned boundary can be nonlinear and thus can capture the characteristics of complicated objectives (e.g., the contour of $f$) quite well, yielding efficient partitioning. **(d) vs evolutionary algorithm (EA)**. CMA-ES generates new samples around the influential mean, which may trap in a local optimum.

**Comparison with gradient-based approaches**: Table 2 summarizes the sample efficiency of SOTA gradient-based approach on 6 MuJoCo tasks. Note that given the prior knowledge that a gradient-based approach (i.e., exploitation-only) works well in these tasks, LA-MCTS, as a black-box optimizer, will spend extra samples for exploration and is expected to be less sample-efficient than the gradient-based approach for the same performance. Despite that, on simple tasks such as *swimmer*, LA-MCTS still shows superior sample efficiency than NG and TRPO, and is comparable to ARS. For high-dimensional tasks, exploration bears an excessive burden and LA-MCTS is not as sample-efficient as other gradient-based methods in MuJoCo tasks. We leave further improvement for future work.

**Comparison with LaNAS**: LaNAS lacks a surrogate model to inform sampling, while LA-MCTS samples with BO. Besides, the linear boundary in LaNAS is less adaptive to the nonlinear boundary used in LA-MCTS (e.g. Fig. 8(b)).

## 4.2 Small-scale Benchmarks

The setup of each methods can be found at Sec A.1 in appendix, and figures are in Appendix A.2.

**Synthetic functions**: We further benchmark with four synthetic functions, Rosenbrock, Levy, Ackley and Rastrigin. Rosenbrock and Levy have a long and flat valley including global optima, making optimization hard. Ackley and Rastrigin function have many local optima. Fig. 9 in Appendix shows the full evaluations to baselines on the 4 functions at 20 and 100 dimensions, respectively. The result shows the performance of each solvers varies a lot w.r.t functions. CMA-ES and TuRBO work well on Ackley, while Dual Annealing is the best on Rosenbrock. However, LA-MCTS consistently improves TuRBO on both functions.

**Lunar Landing**: the task is to learn a policy for the lunar landing environment implemented in the Open AI gym [59], and we used the same heuristic policy from TuRBO [2] that has 12 parameters to optimize. The state vector contains position, orientation and their time derivatives, and the state of

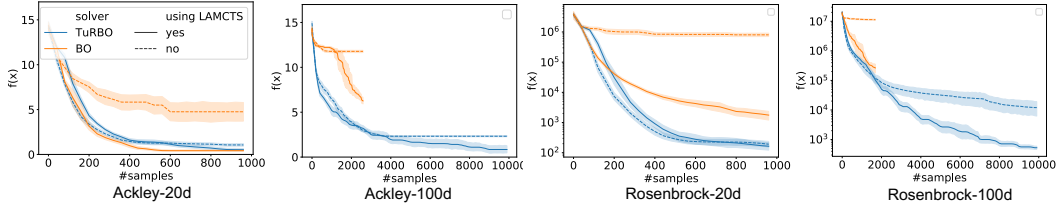

Figure 6: **LA-MCTS as an effective meta-algorithm**. LA-MCTS consistently improves the performance of TuRBO and BO, in particular in high-dimensional cases. We only plot part of the curve (each runs lasts for 3 day) for BO since it runs very slow in high-dimensional space.

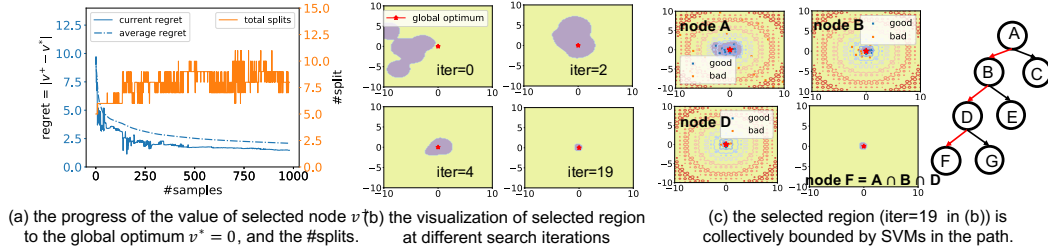

(a) the progress of the value of selected node $v^+$ to the global optimum $v^* = 0$, and the #splits.

(b) the visualization of selected region at different search iterations

(c) the selected region (iter=19 in (b)) is collectively bounded by SVMs in the path.

Figure 7: **Validation of LA-MCTS**: (a) the value of selected node becomes closer to the global optimum as #splits increases. (b) the visualization of $\Omega_{selected}$ in the progress of search. (c) the visualization of $\Omega_{selected}$ that takes the intersection of nodes on the selected path.

being in contact with the land or not. The available actions are firing engine left, right, up, or idling. Fig. 10 shows LA-MCTS performs the best among baselines.

**Rover-60d**: the task was proposed in [18] that optimizes 30 coordinates in a trajectory on a 2d plane, so the state vector consists of 60 variables. LA-MCTS still performs the best on this task.

## 4.3 Validation of LAMCTS

**LA-MCTS as an effective meta-algorithm**: LA-MCTS internally uses TuRBO to pick promising samples from a sub-region. We also try using regular Bayesian Optimization (BO), which utilizes Expected Improvement (EI) for picking the next sample to evaluate. Fig. 6 shows LA-MCTS successfully boosts the performance of TuRBO and BO on Ackley and Rosenbrock function, in particular for high dimensional tasks. This is consistent with our results in MuJoCo tasks (Fig. 5).

**Validating LA-MCTS**. Starting from the entire search space $\Omega$, the node model in LA-MCTS recursively splits $\Omega$ into a high-performing and a low-performing regions. The value of a region $v^+$ is expected to become closer to the global optimum $v^*$ with more and more splits. To validate this behavior, we setup LA-MCTS on Ackley-20d in the range of $[-5, 10]^{20}$, and keeps track of the value of a selected partition, $v_i^+ = \frac{1}{n_i} \sum f(\mathbf{x}_i), \forall \mathbf{x}_i \in D_t \cap \Omega_{selected}$, and as well as the number of splits at each steps. The global optimum of Ackley is at $v^* = 0$. We plot the progress of regret $|v_i^+ - v^*|$ in the left axis of Fig. 7(a), and the number of splits in the right axis of Fig. 7(a). Fig. 7 shows the regret decreases as the number of splits increases, which is consistent with the expected behavior. Besides, spikes in the regret curve indicate the exploration of less promising regions from MCTS.

**Visualizing the space partition**. We further understand LA-MCTS by visualizing space partition inside LA-MCTS on 2d-Ackley in the search range of $[-10, 10]^2$, which the global optimum $v^*$ is marked by a red star at $\mathbf{x}^* = \mathbf{0}$. First, we visualize the $\Omega_{selected}$ in first 20 iterations, and show them in Fig. 7(b) and the full plot in Fig. 11(b) at Appendix. The purple indicates a good-performing region, while the yellow indicates a low-performing region. In iteration = 0, $\Omega_{selected}$ misses $v^*$ due to the random initialization, but LA-MCTS consistently catches $v^*$ in $\Omega_{selected}$ afterwards. The size of $\Omega_{selected}$ becomes smaller as #splits increases along the search (Fig. 7(a)). Fig. 7(c) shows the selected region is collectively bounded by SVMs on the path, i.e. $\Omega_F = \Omega_A \cap \Omega_B \cap \Omega_D \cap \Omega_F$.

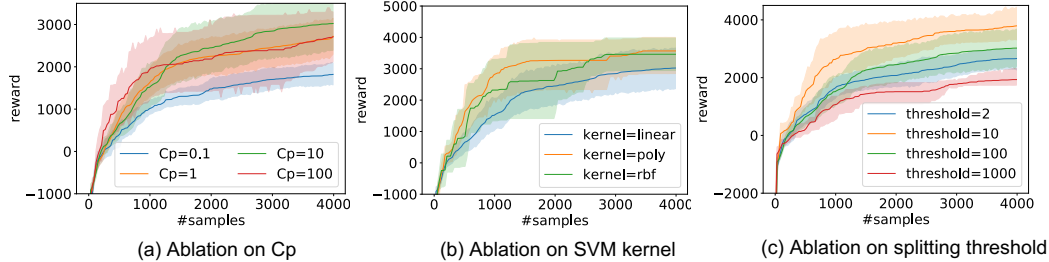

Figure 8: Ablation studies on hyper-parameters of LAMCTS.

### 4.4 Ablations on hyper-parameters

Multiple hyper-parameters in LA-MCTS, including $C_p$ in UCB, the kernel type of SVM, and the splitting threshold ($\theta$), could impact its performance. Here ablation studies on *HalfCheetah* are provided for practical guidance.

**Cp**: $C_p$ controls the amount of exploration. A large $C_p$ encourages LA-MCTS to visit bad regions more often (exploration). As shown in Fig 8, too small $C_p$ leads to the worst performance, highlighting the importance of exploration. However, a large $C_p$ leads to over-exploration which is also undesired. We recommend setting $C_p$ to 10% to 1% of max $f(\mathbf{x})$.

**The SVM kernel**: the kernel type decides the shape of the boundary drawn by each SVM. The linear boundary yields a convex polytope, while polynomial and RBF kernel can generate arbitrary region boundary, due to their non-linearity, which leads to better performance (Fig 8(b)).

**The splitting threshold** $\theta$: the splitting threshold controls the speed of tree growth. Given the same #samples, smaller $\theta$ leads to a deeper tree. If $\Omega$ is very large, more splits enable LA-MCTS to quickly focus on a small promising region, and yields good results ($\theta = 10$). However, if $\theta$ is too small, the performance and the boundary estimation of the region become more unreliable, resulting in performance deterioration ($\theta = 2$, in Fig. 8).

## 5 Conclusion and future research

The global optimization of high-dimensional black-box functions is an important topic that potentially impacts a broad spectrum of applications. We propose a novel meta method LA-MCTS that learns to partition the search space for Bayesian Optimization so that it can attend on a promising region to avoid over-exploring. Comprehensive evaluations show LA-MCTS is an effective meta-method to improve BO. In the future, we plan to extend the idea of space partitioning into Multi-Objective Optimizations.

## 6 Broader impact

Black-box optimization has a variety of applications in practice, ranging from the hyper-parameter tuning in the distributed system and database, Integrated Circuit(IC) design, Reinforcement Learning (RL), and many more. Most real-world problems are heterogeneous and high-dimensional while existing black-box solvers struggle to yield a reasonable performance in these problems. In this paper, we made our first step to show a gradient-free algorithm partially solves high-dimensional complex MuJoCo tasks, indicating its potential to other high-dimensional tasks in various domains.

Switching to LA-MCTS may improve the productivity at a minimal cost when searching for better performance in a wide variety of applications where the gradient of the function is not known. At the same time, we don't foresee any negative social consequences.

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
