[Supplementary Material]

# A  Additional Experiment Details

## A.1  Hyper-parameter settings for all baselines in benchmarks

**Setup for MuJoCo tasks**: Fig. 5 shows 13 methods in total, and here we describe the hyper-parameters of each method. Since we're interested in the sample efficiency, the batch size of every method is set to 1. We reuse the policy and evaluation codes from ARS [57], and the URL to the ARS implementation can be found in the bibliography. The implementations of VOO, SOO, and DOO are from here [1]; methods including CMA-ES, Differential Evolution, Dual Annealing are from the optimize module in scipy, and Shiwa is from Nevergrad[2]. Please see the bibliography for the reference implementations of BOHB and HesBO.

LA-MCTS    we use 30 samples for the initialization; and the SVM uses RBF kernel for easy tasks including *swimmer*, *hopper*, *half-cheetah*, and linear kernel for hard tasks including *2d-walker*, *ant* and *humanoid* to get over $3 * 10^4$ samples. $C_p$ is set to 10, and the splitting threshold $\theta$ is set to 100. LA-MCTS uses TuRBO-1 for sampling, and the setup of TuRBO-1 is exactly the same as TuRBO described below. TuRBO-1 returns all the samples and their evaluations to LA-MCTS once it hits the re-start criterion.

TuRBO    we use 30 samples for the initialization, and CUDA is enabled. The rest hyper-parameters use the default value in TuRBO. In MuJoCo, we used TuRBO-20 that uses 20 independent trust regions for the best $f(x)$.

LaNAS    we use 30 samples for the initialization; the height of search tree is 8, and $C_p$ is set to 10.

VOO    default setting in the reference implementation.

DOO    default setting in the reference implementation.

SOO    default setting in the reference implementation.

CMA-ES    the initial standard deviation is set to 0.5, and the rest parameters are default in Scipy.

Diff-Evo    default settings in Scipy.

Shiwa    default settings in Nevergrad.

Annealing    default settings in Scipy.

BOHB    default settings in the reference implementation.

HesBO    The tuned embedding dimensions for *swimmer*, *hopper*, *walker*, *half-cheetah*, *ant*, and *humanoid* are 8, 17, 50, 50, 400, and 1000, respectively.

**Setup for synthetic functions, lunar landing, and trajectory optimization**: similar to MuJoCo tasks, the batch size of each method is set to 1. The settings of VOO, DOO, SOO, CMA-ES, Diff-Evo, Dual Annealing, Shiwa, BOHB, TuRBO are the same as the settings from MuJoCo. We modify $C_p = 1$ and the splitting threshold $\theta = 20$ in LA-MCTS. Similarly, we also changed $C_p = 1$ in LaNAS. We set the upper and lower limits of each dimension in Ackley as [-5, 10], Rosenbrock is set within [-10, 10], Rastrigin is set within [-5.12, 5.12], Levy is set within [-10, 10].

**Runtime**: LaNAS, VOO, DOO, SOO, CMA-ES, Diff-Evo, Shiwa, Annealing are fairly fast, which can collect thousands of samples in minutes. The runtime performance of LAMCTS and TuRBO are consistent with the result in [2] (see sec.G in appendix) that collects $10^4$ samples in an hour using 1 V100 GPU. BOHB and HesBO toke up to a day to collect $10^4$ samples for running on CPU.

## A.2   Additional experiment results

Figure 9: **Evaluations on synthetic functions**: the best method varies w.r.t functions, while LA-MCTS consistently improves TuRBO and being among top methods among all functions.

(a) Lunar landing, #params = 12    (b) Rover trajectory planning, #params = 60

Figure 10: **Evaluations on Lunar landing and Trajectory Optimization**: LA-MCTS consistently outperforms baselines.

Figure 11: **The visualization of LA-MCTS in iterations 1→20**: the purple region is the selected region $\Omega_{selected}$, and the red star represents the global optimum.

## Footnotes

[1]https://github.com/beomjoonkim/voot

[2]https://github.com/facebookresearch/nevergrad