[Reviews · NeurIPS 2020]

Review 1

Summary and Contributions: The authors propose a meta-algorithm to learn promising regions of the search space in high-dimensional optimization problems. The benefit of the presented framework is that it can easily be used in combination with existing methods, e.g., Bayesian Optimization, to scale these to high-dimensional search spaces. The main contribution is to use MCTS to learn promising regions--or in other words to partition the search space--to avoid over-exploration.

Strengths: The paper considers an interesting problem that has seen a lot of attention in recent years. - Soundness of the claims in terms of empirical evaluation: The presented algorithm has been evaluated extensively on a wide range of test functions and RL problems. The comparison to other state of the art methods is impressive. Further, the comparison to gradient-based approaches on the RL tasks is interesting. The additional visualization and analysis in Figure 5 helps to understand how the algorithms works in practice. - Significance and novelty: The idea of learning interesting regions of the search space is appealing and seems to work well for the problems evaluated in the paper. Focusing only on local regions in the search space is not novel in itself, see e.g., TuRBO. Using MCTS in combination with BO is (apart from LaNAS) a novel contribution. - Relevance to the community: This work is in line with many other papers presented at top tier ML venues such as NeurIPS and ICML and the community would certainly benefit from the general idea proposed in this paper.

Weaknesses: While the general idea of the paper is appealing and has been evaluated extensively, the presentation of the methodology is lacking in clarity at times. After reading section 3, some issues could have been addressed more clearly: • Regarding line 171/172: what do the authors mean by “regret reaches the plateau”? Can this be quantified? • If the tree is constructed as described, it is questionable that the leftmost leave is actually the ‘best’ leave. Can this be shown? • How is the problem treated that SVM can lead to many distinct areas in the described methodology? Consider the case of the 1D sine function and we have data points only at increments of pi. K-means would result in two clusters, i.e., the points with values +1 and -1, respectively. Then, SVM would potentially cluster the domain in the two classes resulting in alternating regions for each class. What would be the resulting domain for TuRBO then? In the very beginning, the authors mention that only deterministic objective functions are considered. It is not clear how this statement fits to the the main result: the optimization of policies on the MuJoCo tasks which are known to be inherently stochastic objectives. Does LA-MCTS depend on the deterministic assumption? If yes: why does it work well in practice on stochastic functions, and if not: Why assume it then? The empirical performance of LA-MCTS is impressive. However, the method combines many different building blocks and as such introduces many additional hyperparameters. Though an ablation study was performed, the performance of the method depends drastically on the choice of hyperparameters. As such, the practicability of the approach is limited as an additional layer of parameters needs to be tuned in addition to the BO parameters. It is not exactly clear, why the authors call their method ‘latent actions’ as these are just the decision boundaries from the SVM classifier. No theoretical work is presented. ------------------------------ After reading the authors response: Thank you for the detailed response to the raised concerns as well as the additional experiments. Tree construction: Being the best node in expectation is something different then being the best node. This should be made more clear in the main paper. Further, Figure 10 does not really help to make this issue more clear as for example the evaluated points are missing in the plot. How's the initial purple region selected when no data is available? Also, using a contour-plot to visualize the objective function would help to understand the figure better. Deterministic assumption: if no component depends on being deterministic, than I'd highly recommend removing this from the main paper in the beginning. Also, just using a sample mean of 5 rollouts does not lead to a deterministic function but just reduces the variance by a factor of 5, which can still be relatively high for RL tasks especially as the outcome does not necessarily follow a uni-modal distribution. Also, please make the use of multiple rollouts more transparent as this simplifies the RL problem drastically. Minor: Appendix A.1: Hit-and-Run and Gibbs sampling do not require the region to be a convex polytope. Overall: The approach presented in this paper shows great potential but the quality of the paper is not yet at the level of a top-tier conference.

Correctness: The methods have been tested on a range of standard benchmark functions that are typically used in the community. The methodology seems to be correct overall. The numbers for ARS, NG, TRPO presented in Table 2 seem to be copied from [54], however, a different version of MuJoCo has been used. Line 76: “BOHB [23] further combines TPE with Hyperband [24] to achieve strong any time performance. Therefore, we choose TPE in comparison.” This sounds like TPE is used in the numerical experiments, however, results for BOHB are presented in Section 4, though.

Clarity: The paper is a bit hard to read at times due to grammatical errors and unclear formulations. To name only a few examples: Line 6: … recent works like LaNAS shows good … Line 13: … local model fits well with … Line 26: … the optimizer needs to search every x to find the optimal. Line 38: … K-ary uniform partitions Line 58: … use existing function approximator like BO to find a promising … (BO is not a function approximator) Line 77: Therefore, we choose TPE in comparison. Line 117: This section describes LA-MCTS that progressively learn and generalize … Line 119: … to Table. 1 for … Line 64, 219, 223, 224: switching between different capitalization of MuJoCo. Line 153: Please note the simplicity is critical to the node model for having a tree of them. Figure 4: (collected from runs lasting for 3 day)

Relation to Prior Work: The overall relation to prior work is discussed and all important papers in the field are cited. The general idea seem to stem from the LaNAS algorithm which uses only linear decision boundaries instead of nonlinear. The authors make it clear how LA-MCTS is different from LaNAS.

Reproducibility: Yes

Additional Feedback: In general, please spell-check your paper before submission. Some questions have been raised in the Weaknesses section.


Review 2

Summary and Contributions: LA-MCTS is an MCTS algorithm based on search space partitioning. It adaptively partitions the search space using Kmeans and SVM. It seems to be a simple and effective approach for domains where it is possible to obtain a large number of samples quickly, and the dimension is not so high.

Strengths: The idea is simple and well described. The experimental results are adequate. This paper analyzes both the strength and limitations of their algorithm. I highly appreciate it because most papers hide their limitations. At first glance, I worried that the expansion threshold \theta might be too large, but the results in Fig. 6 (c) show that the best value was $\theta = 10$. Therefore it is not so large. It is similar to the expansion threshold used in normal MCTS for games (FYI, AlphaGo [42] used 40).

Weaknesses: Works well in some domains but not for others. However, the paper describes its own weaknesses, which would be very useful for the readers.

Correctness: It seemed mostly correct. The questions and concerns are described in questions for the authors. The comparison with other algorithms in the experiments seems adequate.

Clarity: The explanation of the algorithm was easy to understand and well described with the help of beautiful figures. I had several small concerns, which I wrote in the feedback.

Relation to Prior Work: This paper covers a broad range of related work in recent black-box optimization. The novelty of the proposed algorithms is clear.

Reproducibility: Yes

Additional Feedback: Misc. comments. - Lines in Figure 3 were hard to distinguish. - Where are the references in the supplementary? Some awkward questions - Is this a maximizing problem or a minimizing problem? Is it correct that min f(x) is sampled and higher f(x) is the better region? - Does STOA mean SoTA? - Kmean should be Kmeans? - What does the citation [54] at line 225 mean? - Does "gibbson sampling [58]" (Appendix A) mean "Gibbs sampling"? (comments after rebuttal) Additional experiments and explanations would improve the readability of this paper. However, my main concern with this paper is not the content, but the writing. I still have some concerns about that.


Review 3

Summary and Contributions: This paper extends the LaNAS method to LaMCTS for scalable black-box optimization. It makes a few improvements including a nonlinear classifier to partition the action space and using BO or TuRBO for sampling in a selected region. It conducts extensive comparison with baselines from BO, EA, MCTS and gradient-based methods on Mujoco locomotion tasks and a few small-scale benchmarks. The experiments show the competitive performance as a black-box optimization method.

Strengths: While the main algorithm is a moderate modification from LaNAS, the two main modifications are good choices in order to scalar to high-dimensional state space for black-box optimization. Thorough empirical comparison with various baselines show a convincingly competitive performance especially in high-dimensional space. The ablation study is very helpful to understand the choice of hyper-parameters.

Weaknesses: One concern I have is about the choice of hyper-parameters. It is important to have a robust setting for a practical black-box optimization algorithm at the absence the prior knowledge about a problem. While the authors argue the performance is not sensitive to the hyper-parameters, the ablation study does show the impact of those parameters. In the experiments, most baselines use their default hyper-parameters for all experiments but C_p, \theta and the type of kernel of LaMCTS are adjusted according to tasks. It makes me wonder if LaMCTS can maintain its performance with a single setting. Besides, how is the length-scale of the RBF kernel decided?

Correctness: The method is correctly presented. The empirical evaluation is correct, except for the choice of hyper-parameters as explained in the "weakness" question.

Clarity: The paper is well written. Each component of the whole algorithm is clearly and concisely explained in section 3 despite the limit of space.

Relation to Prior Work: The authors have done a great job explaining the relation of the proposed algorithm to the work of LaNAS, Bayesian optimization, evolutionary algorithms and MCTS.

Reproducibility: Yes

Additional Feedback: Typo: "STOA" -> "SOTA"

[Author Response · NeurIPS 2020]



Rebuttal-Fig. 1: the visualization of partitioning 1d $\sin(x)$.

**(a) search tree**

**(b) split on node a**
- average sin(x): -0.234
- average sin(x): 0.127

**(c) split on node b**
- average sin(x): 0.05
- average sin(x): 0.204

**(d) split on node d**
- average sin(x): -0.123
- average sin(x): 0.694

**(e) split on node f**
- average sin(x): 0.398
- average sin(x): 0.99

Rebuttal-Fig. 2: LA-MCTS on Walker2d

| Task | Reward Threshold | #episodes needed by LA-MCTS to get threshold |
|---|---|---|
| Swimmer-v1 | 325 | **126** |
| Hopper-v1 | 3120 | 2913 |
| HalfCheetah-v1 | 3430 | 3967 |
| Walker2d-v1 | 4390 | N/A($r_{best} = 3523$) |
| Ant-v1 | 3580 | N/A($r_{best} = 2871$) |
| Humanoid-v1 | 6000 | N/A($r_{best} = 3202$) |

Table 1: Averaged samples to reach the reward threshold on Mujoco-V1. Table. 2 in the main paper uses Mujoco-V2.

We sincerely thank reviewers R1 , R2 , R3 for their constructive feedbacks. We answer the questions as follows:

**R1 Mujoco versions**: Thanks for pointing out! We redo the experiment on Mujoco-V1 in Table. 1. LA-MCTS shows similar performance between V1 and V2 except for Walker2d, where LA-MCTS does slightly better (Rebuttal-Fig.2), consistent with previous reports[1].

**R1 How to quantify regret reaching the plateau**: For each node, a minimal number of samples are needed to establish a decent local model, while more samples do not help improve its performance substantially, due to the curse of dimensionality. This is when *a plateau of regret* happens. In this case, it is better to split the region so that the future sampling focuses on promising (and smaller) regions, yielding higher sample efficiency. To determine when to split, we introduced a hyper-parameter *splitting threshold* and provided ablation study in Fig. 6(c), which indeed shows there is a sweetspot of splitting threshold. We will clarify it in the paper.

**R1 Partition of one-dimensional $\sin(x)$. Will interleaving high/low function values cause problem?**: We cluster data points using $(\mathbf{x}, f(\mathbf{x}))$, which are $d + 1$ dimensional vectors. Since it involves the input features $\mathbf{x}$, K-means will consider the vicinity of data points and group close points together, preventing the interleaving pattern from happening. As shown in Rebuttal-Fig. 1, for one-dimensional $\sin(x)$, for the splitting sequence $a \rightarrow b \rightarrow d \rightarrow f \rightarrow h$, LA-MCTS first groups local regions together, then gradually focuses on a particular peak and make refinements around it.

**R1 Show the leftmost is the best leaf**: By construction, the value of a left child ($v_l$) > a right child ($v_r$) and by recursively applying this rule on the tree, the leftmost node *is expected to be* the highest value node. However, it is possible that the current local models at each nodes may not be correct, due to insufficient samples. Fig. 10 in appendix shows this behavior (iterations 0→3). The exploration in MCTS alleviates this issue by visiting different leaves to capture a global view of the search space and update the learned partition accordingly.

**R1 Deterministic assumption in LA-MCTS**: LA-MCTS can also be applied to stochastic black-box function. None of its components require the function to be deterministic, while it is possible that for stochastic function, more samples are needed to learn partition and to fit a local model at the leaves. In Mujoco, LA-MCTS, as well as all black-box optimization baselines we compare against, uses an average rewards from 5 different trajectories (or episodes) to mimic deterministic rewards when evaluating a sampled policy.

**R3 Hyperparameters**. *1) How to choose the length scale of RBF?* We used SVM in scikit-learn, and the length scale of a RBF kernel is decided by a hyper-parameter gamma with two choice of values *auto* and *scale*. We notice scale is better than auto in practice. *2) Choice of hyper-parameters for baselines?* We do have carefully chosen the hyper-parameters for baselines. For example, Shiwa is a meta-method that internally optimizes hyper-parameters for CMA-ES; we used suggested hyper-parameters from scikit-learn for Diff-Evo, Anneal and CMA-ES. The setting of TuRBO inside LA-MCTS is exactly the same as TuRBO used in baselines; We also tuned the embedding size for HesBO and used the suggested settings for BOHB.

**R2 Is LA-MCTS maximizing?** LA-MCTS is maximization; and we can change a minimization to a maximization by multiplying $-1$ in $f(\mathbf{x})$.

**Other issues**. We will correct typos in the next iteration. R2 The citation (line 225) is to point out the source of using a linear policy. "gibbson sampling" means "Gibbs sampling" and we will fix all "STOA" to be "SoTA".

## Footnotes

[1] https://github.com/openai/gym/pull/834


[Meta-Review · NeurIPS 2020]

The paper contains some interesting ideas of partitioning the search space in Bayesian optimization. Experimental studies are good, although the paper could benefit from another round of editing for the camera ready. Reviewers were not overly excited about the paper, but also did not identify any fundamental flaws. As such, it is recommended for acceptance as a poster. We strongly encourage the authors to take the feedback from the reviews into account when preparing the camera-ready version.